# Monoclonal Antibodies as an Antibacterial Approach Against Bacterial Pathogens

**DOI:** 10.3390/antibiotics9040155

**Published:** 2020-04-01

**Authors:** Daniel V. Zurawski, Molly K. McLendon

**Affiliations:** Wound Infections Department, Bacterial Diseases Branch, Walter Reed Army Institute of Research, Silver Spring, MD 20910, USA; molly.k.mclendon.ctr@mail.mil

**Keywords:** antibodies, ESKAPE pathogens, *Escherichia coli*, antibacterial, therapeutic, clinical trial

## Abstract

In the beginning of the 21st century, the frequency of antimicrobial resistance (AMR) has reached an apex, where even 4th and 5th generation antibiotics are becoming useless in clinical settings. In turn, patients are suffering from once-curable infections, with increases in morbidity and mortality. The root cause of many of these infections are the ESKAPEE pathogens (*Enterococcus* species, *Staphylococcus aureus*, *Klebsiella pneumoniae*, *Acinetobacter baumannii*, *Pseudomonas aeruginosa*, *Enterobacter* species, and *Escherichia coli*), which thrive in the nosocomial environment and are the bacterial species that have seen the largest rise in the acquisition of antibiotic resistance genes. While traditional small-molecule development still dominates the antibacterial landscape for solutions to AMR, some researchers are now turning to biological approaches as potential game changers. Monoclonal antibodies (mAbs)—more specifically, human monoclonal antibodies (Hu-mAbs)—have been highly pursued in the anti-cancer, autoimmune, and antiviral fields with many success stories, but antibody development for bacterial infection is still just scratching the surface. The untapped potential for Hu-mAbs to be used as a prophylactic or therapeutic treatment for bacterial infection is exciting, as these biologics do not have the same toxicity hurdles of small molecules, could have less resistance as they often target virulence proteins rather than proteins required for survival, and are narrow spectrum (targeting just one pathogenic species), therefore avoiding the disruption of the microbiome. This mini-review will highlight the current antibacterial mAbs approved for patient use, the success stories for mAb development, and new Hu-mAb products in the antibacterial pipeline.

## 1. Introduction

Steve Projan (former Vice President, Head of Infectious Diseases and Vaccines at AstraZeneca/MedImmune) has been one of the most ardent supporters of immunotherapies for bacterial infections over the last two decades. His experience, leadership, and knowledge drove a successful program of more than 30 people at MedImmune, Inc. (now AstraZeneca plc) generating bi-specific antibody solutions against bacterial infections (discussed below) and subsequently led to numerous keynote talks at antibacterial meetings. He famously starts many of his talks with the story of the Iditarod, the great dogsled race held every year in Alaska. The Iditarod began as commemoration of events that occurred in 1925 when a diphtheria infection spread through the town of Nome. Diptheria is caused by the bacterium *Corynebacterium diphtheriae*. At the time, the only solution for this lethal infection was serum that had been isolated from horses injected with diphtheria toxin, a toxin made by *C*. *diphtheriae* [1]. Because of harsh weather conditions, the only way to get the serum to Nome before it expired was by dogsled relay from Nenana, located 674 miles (1085 km) from Nome [2]. Estimates suggest 10,000 lives were saved in Nome and surrounding villages because of the heroic efforts of twenty sled dog teams running the serum across the Alaskan wilderness (Figure 1), in just six days, a journey that would normally take more than twenty days [2]. Horse serum, and the antibodies within, was the means of combatting *C. diphtheriae* infections in this case, but at the time, this method was also used to treat infections caused by bacterial species such as *Streptococcus*, *Neisseria*, and *Haemophilus* [3].

Until the Golden Age of Antibiotics began in the 1940s with the advent of penicillin, delivering passive immunity via horse serum or using bacteriophage therapy were the standards to treat bacterial infection [4,5], and the epic story of the birth of the Iditarod indicates how important serum was at limiting the spread of a bacterial infection outbreak. However, as more antibiotics were discovered and brought to market, it could be argued with good reason that small molecules were the better approach for controlling bacterial infection considering cost and efficacy [5]. Because of this, small molecule-based antibiotics dominated the antibacterial space for the next sixty years and still do. However, with the dawn of multidrug-resistant (MDR) strains and the present day emergence of extensively drug-resistant (XDR) and pandrug-resistant (PDR) strains of the ESKAPEE pathogens (*Enterococcus* species, *Staphylococcus aureus*, *Klebsiella pneumoniae*, *Acinetobacter baumannii*, *Pseudomonas aeruginosa*, *Enterobacter* species, and *Escherichia coli*), it is clear that the Golden Age of Antibiotics is over, and the medical and research communities are seeking alternative solutions to traditional small-molecule antibacterial approaches.

Hybridomas were first discovered in the early 1970s by Kilner and Milstein [6], who went on to earn the Nobel Prize for this work. Since then, many developments have been made to improve the performance of monoclonal antibodies (mAbs) as therapeutics. The first of these was to show that a human cell line could also be used for the hybridoma process [7]. This early advance was born from the concept that fully human antibodies would make better therapeutics as they would less likely be cleared by the human immune system. Over the next decade, researchers discovered additional methods to further this idea. Key advances included identifying fully functional mouse mAbs and then humanizing them [8,9]. Others realized that isolating mAbs directly from patients who were infected and cleared infections could be important tools for identifying human monoclonal antibodies (Hu-mAbs) to neutralize the infectious agent [10]. Modern approaches have shown that once obtained, these Hu-mAbs can be sequenced, generated recombinantly, and generated in large quantities for clinical use [11]. More recent advances also include the development of phage display libraries of engineered Hu-mAbs that can dramatically increase the total number of human antibodies for testing to increase the likelihood of finding a unique antibody to interfere with or delay a disease-state [12]. Molecular biology and recombinant DNA techniques have also allowed researchers to modify different amino acids of the antibody’s structure to improve stability and utility in the host [13]. On the whole, the state of the science has advanced to where three major approaches are typically used to isolate/generate Hu-mAbs for therapeutic use: combinatorial display libraries, humanized mice, or single B cell cloning, and each of these strategies are thoroughly discussed in the review by Walter and Burton [14]. Additionally, others are using adenovirus [15,16] or recombinant DNA [17] to produce antibodies in the body, essentially a vaccination-like approach where the viral-encoded DNA or the naked DNA is then translated into a human antibody. Thus, rather than injecting the Hu-mAbs, the human body essentially becomes a Hu-mAb factory, continuously generating the antibody over a period of time. New data suggest that the recombinant DNA platform approach could also be used for bacterial infections as was shown recently in animals [18]. Nonetheless, the driving agent of protection is the Hu-mAb, whether it is made inside the body or injected intravenously.

Aside from the fact that Hu-mAbs are a human therapeutic product, which, in turn, minimizes toxicity concerns, there are other advantages of Hu-mAbs that make their pursuit a promising antibacterial approach. The first of these is longevity, as Hu-mAbs are not cleared by the host immune system as the half-life is typically 21 days for IgG subtypes [11]. Second, Hu-mAbs used as treatment against one bacterial species confers inherent pathogen specificity that does not disrupt normal bacterial flora in the body. Third, it potentiates both rapid and sustained killing via multiple mechanisms including: direct killing, anti-virulence, neutralization, complement deposition, and opsonization by phagocytes [11,19,20]. Furthermore, mAbs with Fc domains that bind to the host phagocyte receptor FcγRII result in downstream suppression of inflammation and sepsis caused by Gram-negative bacteria [11,21]. Killing bacteria by these multiple mechanisms limits toxic shock and the emergence of resistance. Further, tapping into the full capacity of the immune system allows for a diverse repertoire of cell types and killing machinery to clear bacteria from various locales in the body. It should also be noted that small molecules alone never completely clear bacteria; these chemicals will always require the immune system to help clear the remaining bacteria and infection. Some of the ways mAbs can disrupt bacterial function and survival are presented as examples in Table 1 below, and some of this knowledge originally comes from vaccine-based approaches.

## 2. Antibacterial Antibodies—Previous Success

From 2002, when the first fully human antibody was approved (HUMIRA^®^/adalimumab), to 2016, over 40 Hu-mAbs were approved by the United States Food and Drug Administration (FDA) for different diseases and treatments [11]. Most of these are related to cancer and autoimmune disease, and none were for bacterial infection. Of note, with respect to infectious disease, SYNAGIS^®^ (palivizumab) was approved for respiratory syncytial virus RSV in 2004 [36]. In 2016, the first Hu-mAb for antibacterial treatment was approved by the FDA: Anthim^®^ (obiltoxaximab), an injection to treat inhalational anthrax in combination with appropriate antibiotics (often ciprofloxacin). Anthim^®^ was also approved to prevent inhalational anthrax when alternative therapies are not available or not appropriate via the Animal Rule for biothreat organisms [37]. The Animal Rule, put in place by the FDA in 2002, allows for the approval of a drug for biothreat organisms that, if untreated, leads to death or serious disability; therefore, safety and efficacy is based on the results in animal models that best represent the clinical indication being targeted [38]. Using the Animal Rule, efficacy of Anthim^®^ was evaluated using New Zealand white rabbits [39]. Inhalation of *Bacillus anthracis* spores cause anthrax infection and, because of the ability of the spores to withstand harsh environments, spores can serve as a source for subsequent infection [40]. Another interesting study, performed after approval, showed that Anthim^®^ could also prevent these types of infections, which further bolstered the product’s utility [40]. Later that year, in October of 2016, the FDA approved the second antibody for bacterial infection, Zinplava™ (bezlotoxumab), for *Clostridum difficile* infections in adults [41]. It is important to understand that this product does not protect from or treat initial or primary *C*. *difficile* infection, rather it was approved to reduce the recurrence of infection, which is often seen with *C*. *difficile* [41,42]. However, unlike Anthim^®^, which relied on just one main study in animals and the Animal Rule, Zinplava™ showed both safety and efficacy in multiple animal models [42,43]. FDA approval then followed the traditional path to approval including Phase 1, Phase 2, and Phase 3 trials [41].

## 3. Previous Failures Lead to Current Success

With the clinical success of Hu-mAbs for other diseases, it is surprising there are not more Hu-mAbs being made for bacterial infections. It is important to understand that research and production of antibodies can be expensive, and there have been some failures in the recent past that may have dampened the enthusiasm of researchers and investors that were considering Hu-mAbs for antibacterial development. One example is KB001-A, a mAb developed by KaloBios for *P*. *aeruginosa* infections. This product was made against the Type III secretion system (T3SS), which is required for *P*. *aeruginosa* pathogenesis [44]. Even though there were positive data in animal models [44,45] and the mAb was found to be safe in human patients [45], the product was not effective for patients with mechanical ventilation in a Phase 2 trial. Unfortunately, there have also been numerous failures with Hu-mAb treatments being developed for *S*. *aureus,* such as tefibazumab, which caused developers to question the monoclonal antibody approach for this bacterial species [46]. These antibodies showed promise in animals, but subsequently failed in Phase 2 efficacy trials [46]. However, there are lessons to be learned from these failures. Using just one antibody against one target, when bacteria secrete or have more than 200-400+ targets on their surface with presumed roles in virulence, may not be sufficient. Further, most approaches do not take into account that bacteria have different lifestyles, and therefore variable surface or secreted protein expression profiles, while residing within the host: vegetative, encapsulated or unencapsulated, biofilm associated, and intracellular, among others [46,47,48,49]. Finally, with respect to the mAb KB-001A, a diagnostic assay was not being used to properly identify patients with *P*. *aeruginosa* at the outset of the study. Because other bacterial species can also be the cause of pneumonia and sometimes infections are polymicrobial, KB-001A was only going to be effective with a subpopulation of patients making it difficult to achieve the threshold for success. However, examples below will highlight how companies are now addressing some of these issues like growth state and the inclusion of an onboard diagnostic with the clinical trial, which can lead to success.

Currently, there are 14 Hu-mAb products in development for nosocomial bacterial pathogens (Table 2). This list is the result of literature searches and publicly available company information; however, it is not exhaustive, as some Hu-mAb initiatives may have been unintentionally overlooked. The focus of the field has mainly been on *S*. *aureus* and *P*. *aeruginosa*, the most prevalent causes of disease with respect to the ESKAPEE pathogens in the Western world (making this more commercially viable), but some newer preclinical approaches have also targeted the difficult-to-treat species such as *C*. *difficile* and *A*. *baumannii*. All but one of the products target individual pathogens, making them narrow spectrum. The exception is the Hu-mAb being developed for biofilm by Trellis. This antibody targets DNABII, which is conserved amongst multiple bacteria and is required for biofilm formation as bacteria often release DNA into the extracellular milieu when establishing a biofilm [24]. If this product shows success in human trials, it could be very exciting to partner it with antibiotics that are often stymied by biofilms.

Of the Hu-mAbs being clinically evaluated, the products from AstraZeneca PLC (formerly MedImmune) have been extensively evaluated in a number of preclinical models. MEDI4893 was developed for *S*. *aureus* and has been shown to be protective in multiple models of pneumonia including ferret, rabbit and mouse [50,51]. The antibody targets the secreted alpha-toxin and prevents the bacterium’s ability to cause apoptosis in cells, which, in turn, prevents lysis and tissue necrosis caused by *S*. *aureus* infection. Further, the antibody has been shown to prevent necrosis in other clinical indications such as surgical site and wound infection models [52,53,54]. Recently, MEDI4893 was shown to improve lung function in patients with *S*. *aureus* infections in a successful Phase 2 trial, and the results were presented earlier this year (ASM Microbe, 2019, San Francisco). The trial design was relatively simple, where the product was administered intravenously (i.v.) at 2000 or 5000 mg on the first day of enrollment, and was compared to a placebo control. Patients were excluded if they were given antibiotics, and only enrolled if positive for *S*. *aureus.* The company is now looking to out-license the product as AstraZeneca PLC has stopped actively resourcing its antibacterial Hu-mAb group. 

Aridis Pharmaceuticals, Inc. is also targeting *S*. *aureus* and the secreted alpha-toxin with AR-301 (Salvecin^®^) (Table 2), but there are no published data around this product and the Phase 2 efficacy trial was designed as an adjunct therapy to be used with standard-of-care antibiotics. Like MEDI4893, patients were administered antibody i.v. once upon the day of trial enrollment, however the dose was 20 mg/kg. AR-301 was also successful in a Phase 2 trial for hospital-acquired pneumonia (HAP)/ventilator-associated pneumonia (VAP), and patients were given a 20 mg/kg dose i.v. upon trial enrollment. According to the company, “Patients treated with AR-301 consistently demonstrated less time spent under mechanical ventilation and higher rates of *S*. *aureus* eradication as compared to those treated with antibiotics alone. AR-301 was deemed to be safe and well tolerated.” AR-301 has been granted Fast Track designation by the FDA and was given an orphan drug designation by the European Medicines Agency (EMA), and the company is currently enrolling patients for a Phase 3 trial according to its website and clinicaltrials.gov The future of both these two *S*. *aureus* products, MEDI4893 and AR-301, could pave the way for other Hu-mAb antibacterial treatments especially with respect to Gram-positive pathogens.

Another product from AstraZeneca PLC (formerly MedImmune) is MEDI3902, which is a bi-specific antibody targeting both the T3SS (PcrV), the same target as the KaloBios product, but in addition, also targets a surface polysaccharide (Psl) [55], which is essential for biofilm formation and virulence of *P*. *aeruginosa* [56,57]. MEDI3902 was successful in a Phase I trial [58] and was evaluated for HAP/VAP infections caused by *P*. *aeruginosa* in a Phase II trial. Like the other Hu-mAb from AstraZeneca PLC, this product was also dosed i.v., excluded patients on antibiotics, and only patients positive for *P*. *aeruginosa* were enrolled. Multiple animal models led to the development of this Hu-mAb [55,59], and aside from the anti-virulence mechanism of action, it was recently shown in vitro that MEDI3902 can steer neutrophils through a “dead zone” to reach and destroy biofilm generated by *P*. *aeruginosa* [60], which highlights another positive aspect of infection remediation.

Aridis also has an anti-*P*. *aeruginosa* product: AR101 (Aerumab™). This product specifically targets the O-antigen of the O11 serotype of lipopolysaccharide (LPS), which the company states “represents 22% of all *P*. *aeruginosa* lung infections”. It was successful in both Phase 1 and Phase 2a trials for HAP/VAP infections. Like AR301, this product was evaluated as adjunct with antibiotics, and has “primary safety endpoints and showed a consistent trend toward improvement in mortality, shorter time to clinical cure of pneumonia, shorter time on mechanical ventilation, and fewer days in the ICU as compared to standard of care antibiotics-alone.” However, this is in stark contrast to Aridis’ AR105 product that failed its Phase 2 trial in September 2019. However, the target of this antibody is alginate, which is required for biofilm formation [61], but it is unclear what other roles this polysaccharide may have with respect to pathogenesis of the organism. 

The future for these *P*. *aeruginosa* products is unclear. As with the other AstraZeneca PLC product, MEDI3902 will likely have to be out-licensed to a large company willing to invest in a Phase 3 trial, or perhaps the Biomedical Advanced Research and Development Authority (BARDA) will support a trial as this government agency has a track record of supporting late-stage antibacterial development efforts. Being specific to just one set of *P*. *aeruginosa* strains, AR101 may be of limited value. While the O11 serotype has been the most virulent and resistant with increased mortality in patients, it may be hard to make the case for limited applicability, as in this recent study, O11 strains only made up 15% of the patients in the trial [62]. In contrast, MEDI3902 recognized PcrV and Psl in the most globally distributed isolates [63]. 

Another interesting approach being pushed forward by Roche Ltd. (formerly Genentech) is RG7861 (anti-*S*. *aureus* TAC, DSTA4637S), which is a THIOMAB™ antibiotic conjugate (TAC) that consists of a human monoclonal antibody directed against a *S*. *aureus* protein conjugated to an antibiotic. Anti-*S*. *aureus* TAC is designed to bind to the surface of *S. aureus* bacteria, thereby putting the antibiotic in close proximity to its target in order to enhance killing. The company also reported positive results against intracellular *S*. *aureus*, which is an important aspect of the bacterium’s pathogenic lifestyle [64]. A Phase I clinical trial is currently evaluating RG7861 for safety. Because this antibody–antibiotic combination is the first of its kind in the clinic, many will be curious of the outcome of the Phase I and subsequent trials [65]. It should also be noted that antibiotic–antibody conjugate chemistry is a difficult endeavor. Development of this product included identifying the best antibiotic to use, identifying the linker between the antibody and antibiotic, and identifying the right place on the antibody for the linkage [66,67]. However, one could certainly argue that the work is worth the effort, as *S*. *aureus* is responsible for thousands of deaths per year, and the product was superior to vancomycin *in vivo* [67].

## 4. Future and Conclusions

The rise of AMR is projected to be a tremendous problem, where deaths resulting from bacterial infection, estimated to be as many as 10 million/year by 2050, will surpass deaths from cancer and heart disease combined [68]. As resistance grows in ESKAPEE pathogens [69], innovative, effective treatments are needed, and Hu-mAbs will be an important part of the solution. It is hoped that the successes discussed in this review and advancement of antibody approaches [14] will stimulate the development of more Hu-mAbs to treat bacterial infections. Although there can be large upfront costs to develop Hu-mAbs against bacteria, the success stories of AstraZeneca/MedImmune and Aridis Pharmaceuticals, Inc. products in clinical trials show that investment in Hu-mAbs can lead to development of effective therapeutics. While some products are being used as a standalone therapeutic or as adjuvants with standard-of-care antibiotics to improve patient outcome, it should be noted that these same products could also be used as a prophylactic to prevent infection. A clinical trial has not been designed yet specifically for this purpose, but in the future, it could certainly be another application for Hu-mAb antibacterial technology. One could envision running a trial for patients who are at risk for surgical site infections that do or do not receive Hu-mAbs along with standard-of-care antibiotics, and then monitoring for development of infection.

One drawback of Hu-mAb development is the cost. A significant amount of research is required upfront to identify the best targets on the bacterial surface and sift through many positive binding antibodies to find the most effective at remediating bacterial infection via different mechanisms of action (Table 1). Once found, scaling up can also be a financial burden, as the primary method is CHO cells, and it can cost $10,000–$20,000 per gram of material [70]. However, improvements in culturing and processing are being made to improve yields, and plant-based production is also possible, which can also improve yield [70]. Given that the cost of an MDR infection and staying in the hospital can run as much as $50,000–$100,000 per patient [68], an investment in Hu-mAbs development appears more reasonable. In relation to small molecules, it should be noted that a study found that the average costs were $29,941 per patient treated with meropenem-vaborbactam or $32,294 with ceftazidime-avibactam [71]. Newer antibiotics will demand a higher cost no matter whether antibody or small molecule. 

In the future, more developmental, preclinical assays will help to serve better antibody design. Critics have long said that Hu-mAbs will still be subject to resistance by bacteria just like antibiotics, but it is important to understand there is a distinct difference. Antibiotics directly kill bacteria, placing genetic pressure on the organism to survive by mutation; survivors, therefore, are resistant to the drug and are left to reestablish infection in the host. However, because antibodies often target virulence factors and not proteins required for survival, a mutation of these targets is likely to make bacteria non-virulent or less virulent, possibly subject to enhanced immune system clearance. A recent study supported this idea and showed that when the alpha-toxin epitope for MEDI4839 was mutagenized, *S*. *aureus* had a reduced fitness cost [72]. Another criticism of this approach is the narrow spectrum of the products, as they largely only target one bacterial species; however, this limitation has been somewhat overcome with the use of onboard diagnostics with clinical trials, where a complement diagnostic antibody or PCR test is used to identify patients infected with a particular bacterial species upfront before treatment. As diagnostics improve, this should facilitate the use of Hu-mAbs as treatments. Further, narrow-spectrum therapeutics hold an advantage in avoiding disrupting the normal microbiome of the patient. Finally, there are some that have said that the efficacy of Hu-mAb therapeutics is not significant enough but, again, this criticism will be addressed in the future as technology improves and more targets are added either via a cocktail or by multivalent antibody engineering. It has already been shown that the addition of just one target with a bispecific antibody improves Hu-mAb efficacy [55,59,60]. Further, there are now pentavalent antibodies and mAb cocktails that are also being explored to improve efficacy [73,74]. One can certainly envision a future, 5–10 years from now, where there could be a Hu-mAb antibody product, cocktail or multivalent molecule, which is developed for each bacterial pathogen that causes difficult-to-treat infections. The antibody mixture would ideally use different mechanisms of action, such as inhibiting bacterial growth or spread while also enhancing clearance by the immune system, (Table 1) thereby putting more pressure on the bacterium. While it is possible that some do not see the need for the large, upfront investment in the defining targets when small molecules are still a cheaper approach, the lack of new drug classes and scaffolds have made finding new small-molecule antibiotics a difficult prospect [5]. In contrast, improvements in antibody technology, coupled with clinical need, could certainly drive more research and funding to the Hu-mAb approach. Ultimately, it could become a faster route than small-molecule development and perhaps revolutionize infectious medicine as was proffered years ago [75]. 

## Figures and Tables

**Figure 1 antibiotics-09-00155-f001:**
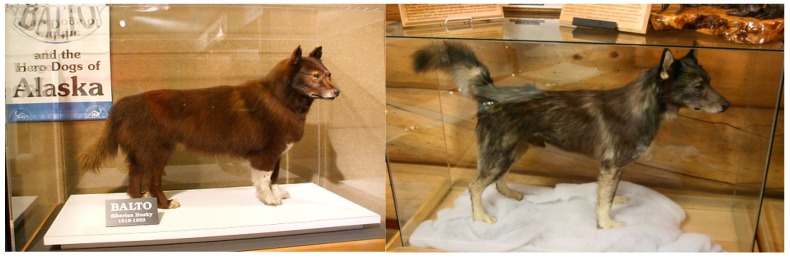
Balto (**left**), one of the heroes of the race to deliver the antitoxin serum to Nome, AK. Upon his death, he was mounted and can still be found at the Cleveland Museum of Natural History in Cleveland, OH. However, Togo (**right**) was the true hero of the relay, running over 250 miles in 3 days. He can be found at the Trail Sled Dog Race Headquarters museum in Wasilla, Alaska.

**Table 1 antibiotics-09-00155-t001:** How antibodies can disrupt bacterial infection.

Antibacterial Mechanism	Reference
Bactericidal	[20,22]
Biofilm	[23,24]
Iron acquisition	[25,26,27]
Attachment/Adhesion	[28,29]
Anti-toxin/Anti-virulence	[30,31]
Opsonophagocytosis	[32,33]
Complement	[34,35]

**Table 2 antibiotics-09-00155-t002:** Companies currently pursuing Hu-mAb therapy for bacterial infections caused by ESKAPEE pathogens and *Clostridum difficile*—products and stage of development.

Name	Bacterial Species Targeted	Company	Development Phase
AR301	*Staphylococcus aureus*	Aridis Pharmaceuticals	Phase 2 Complete Ongoing Phase 3
MEDI4893	*Staphylococcus aureus*	Medimmune	Phase 2 Complete
MEDI3902	*Pseudomonas aeruginosa*	Medimmune	Phase 1 Complete Ongoing Phase 2
AR101	*Pseudomonas aeruginosa*	Aridis Pharmaceuticals	Phase 1 Complete Ongoing Phase 2
514G3	*Staphylococcus aureus*	XBiotech	Phase 2
ARN-100	*Staphylococcus aureus*	Arsansis	Phase 2 Halted
PolyCAb	*Clostridium difficile*	MicroPharm	Phase 1
RG7861	*Staphylococcus aureus*	Roche	Phase 1
TRL1068	Biofilm—multiple species	Trellis Bioscience	Preclinical Entering Phase 1
AR401-mAb	*Acinetobacter baumannii*	Aridis Pharmaceuticals	Preclinical
VXD-003	*Acinetobacter baumannii*	VaxDyn	Preclinical
Cd-ISTAb	*Clostridium difficile*	Integrated BioTherapeutics	Preclinical
ASN-4	*Escherichia coli* (ST131)	Arsansis—Outlicensed to BB100	Preclinical
ASN-5	*K. pneumoniae*	Arsansis—Outlicensed to BB200	Preclinical

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
