# Peer review of "Monoclonal Antibodies as an Antibacterial Approach Against Bacterial Pathogens"

_antibiotics, 2020, doi:10.3390/antibiotics9040155_

Round 1

Reviewer 1 Report

In this review manuscript, the authors provide a potential role of Hu-mABs to be used in prophylactic or therapeutic approaches. This is an important area of ongoing research and this review merits the scope of the journal. The authors also disclosed that there are no conflict of interest or potential partnerships with industries. Overall, this is a comprehensively covered review on the current state of the Hu-mAB technology for anti-bacterial applications. I have not found any errors or pitfalls in the manuscript.

Author Response

Thank you for your positive review.  Our focus will be on the improvements suggested by the other reviewers.

Reviewer 2 Report

Dear Editor in chief:

Antibiotics

Regarding Manuscript: antibiotics-740523

``Monoclonal antibodies as an antibacterial approach against bacterial pathogens`` by Daniel V. Zurawski and Molly K. McLendon:

Authors have described the value of human antibodies targeting different bacteria or pathogens.

 Overall, this manuscript has been written well. However, before considering paper for publication I suggest authors to:

  • Provide a cumulative table showing different methods for targeting bacteria.
  • Another table that showed current approved small molecule inhibitor of bacteria.

Author Response

Thank you for your review.  We have now included a table that includes the different mechanisms of targeting bacteria with an antibody.  The table is essentially a summary of the closing paragraph of the introduction (Lines 95-109) as well as additional mechanisms of blocking bacterial pathogenesis. 

We believe the second table that is being requested is beyond the scope of this work.  We are limiting our review to mAbs, and including a table with small molecules additionally would be a very large table and would take away from our focus of the manuscript.

Reviewer 3 Report

This mini-review article provides a brief overview of the status of antibiotic resistance in ESKAPEE pathogens and makes a case for exploiting the potential of monoclonal antibodies (mAbs) as prophylactic and/or therapeutic strategies to counter the existing and emerging threats posed by nosocomial bacterial infections and the development of resistance to known treatment options. The manuscript is generally well-written and ably engages the reader due to the importance of the topics being discussed and systematic organization of the subheadings allowing for a nice flow of the ideas. Consideration of the following points should enhance further enhance the quality of the overall presentation.

  1. Major: The efficacy of a number of mAbs in preclinical and clinical settings listed in Table 1 is mentioned, but it is felt that the discussion of this aspect would benefit from including the details of how these mAbs were administered, what was the dosing regimen, and the potential mechanism of action of mAbs.
  2. Minor: It would be helpful to mention the monoclonal antibodies as mAbs rather than mAb, which may yield an impression that it is a single purified clone of mAbs.  

Author Response

Major: The efficacy of a number of mAbs in preclinical and clinical settings listed in Table 1 is mentioned, but it is felt that the discussion of this aspect would benefit from including the details of how these mAbs were administered, what was the dosing regimen, and the potential mechanism of action of mAbs.

Thank you for pointing this out.  We now report how the antibodies are administered and dosed in the discussion of each clinically relevant antibody mentioned. The potential mechanisms; however, were already discussed with each antibody product, and now we have also expanded the Introduction by adding an additional table to further highlight potential mechanisms of action.

Minor: It would be helpful to mention the monoclonal antibodies as mAbs rather than mAb, which may yield an impression that it is a single purified clone of mAbs. 

This has now been corrected throughout the manuscript.

Reviewer 4 Report

Well-written article. 

I would like authors to add some discussion on points below,

1. Generation of Antibody is extremely expensive compare to small molecules. From patient view point, it is not a feasible solution. It would be nice if authors could include a paragraph regarding the cost comparison between antibody-Antibiotics vs exiting antibiotics.

2. Challenge of generating Antibody-antibiotic conjugate. 

2. Advantages of targeting more than one toxin using bispecific antibodies and few instances. 

Author Response

  1. Generation of Antibody is extremely expensive compare to small molecules. From patient view point, it is not a feasible solution. It would be nice if authors could include a paragraph regarding the cost comparison between antibody-Antibiotics vs exiting antibiotics.

We have brought up the cost of antibodies showing that while expensive, it is still feasible given that some of the newer small molecule therapies are now costing hospitals $20,000 - $30,000 per patient. This is now discussed in the final paragraph with some new references.

  1. Challenge of generating Antibody-antibiotic conjugate.

Agree that the chemistry of Antibody-antibiotic conjugate is difficult.  This is briefly highlighted now in the last sentence when discussing the Roche/Genentech product now.

  1. Advantages of targeting more than one toxin using bispecific antibodies and few instances.

We previously discussed the one case of MedImmune’s bispecific antibody for multiple targets, but now we include further discussion in the conclusion paragraph (lines 246-249) where we highlight this approach further.